# Visual Object Networks: Image Generation with Disentangled 3D Representation

**Jun-Yan Zhu**
MIT CSAIL

**Zhoutong Zhang**
MIT CSAIL

**Chengkai Zhang**
MIT CSAIL

**Jiajun Wu**
MIT CSAIL

**Antonio Torralba**
MIT CSAIL

**Joshua B. Tenenbaum**
MIT CSAIL

**William T. Freeman**
MIT CSAIL, Google

## Abstract

Recent progress in deep generative models has led to tremendous breakthroughs in image generation. However, while existing models can synthesize photorealistic images, they lack an understanding of our underlying 3D world. We present a new generative model, *Visual Object Networks (VON)*, synthesizing natural images of objects with a disentangled 3D representation. Inspired by classic graphics rendering pipelines, we unravel our image formation process into three conditionally independent factors—shape, viewpoint, and texture—and present an end-to-end adversarial learning framework that jointly models 3D shapes and 2D images. Our model first learns to synthesize 3D shapes that are indistinguishable from real shapes. It then renders the object's 2.5D sketches (i.e., silhouette and depth map) from its shape under a sampled viewpoint. Finally, it learns to add realistic texture to these 2.5D sketches to generate natural images. The VON not only generates images that are more realistic than state-of-the-art 2D image synthesis methods, but also enables many 3D operations such as changing the viewpoint of a generated image, editing of shape and texture, linear interpolation in texture and shape space, and transferring appearance across different objects and viewpoints.

## 1 Introduction

Modern deep generative models learn to synthesize realistic images. Figure 1a shows several cars generated by a recent model [Gulrajani et al., 2017]. However, most methods have only focused on generating images in 2D, ignoring the 3D nature of the world. As a result, they are unable to answer some questions that would be effortless for a human, for example: what will a car look like from a different angle? What if we apply its texture to a truck? Can we mix different 3D designs? Therefore, a 2D-only perspective inevitably limits a model's practical application in fields such as robotics, virtual reality, and gaming.

In this paper, we present an end-to-end generative model that jointly synthesizes 3D shapes and 2D images via a disentangled object representation. Specifically, we decompose our image generation model into three conditionally independent factors: shape, viewpoint, and texture, borrowing ideas from classic graphics rendering engines [Kajiya, 1986]. Our model first learns to synthesize 3D shapes that are indistinguishable from real shapes. It then computes its 2.5D sketches [Barrow and Tenenbaum, 1978, Marr, 1982] with a differentiable projection module from a sampled viewpoint. Finally, it learns to add diverse, realistic texture to 2.5D sketches and produce 2D images that are indistinguishable from real photos. We call our model Visual Object Networks (VON).

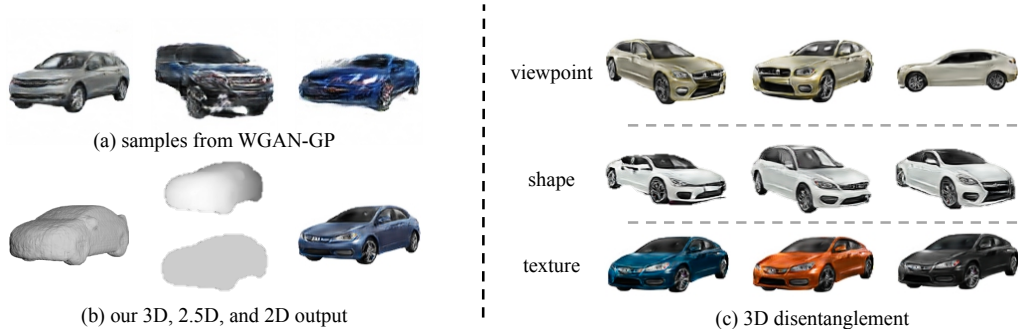

(a) samples from WGAN-GP

(b) our 3D, 2.5D, and 2D output

viewpoint

shape

texture

(c) 3D disentanglement

Figure 1: Previous 2D GANs vs. Visual Object Networks (VON). (a) Typical examples produced by a recent GAN model [Gulrajani et al., 2017]. (b) Our model produces three outputs: a 3D shape, its 2.5D projection given a viewpoint, and a final image with realistic texture. (c) Given this disentangled 3D representation, our method allows several 3D applications including changing viewpoint and editing shape or texture independently. Please see our code and website for more details.

Wiring in conditional independence reduces our need for densely annotated data: unlike classic morphable face models [Blanz and Vetter, 1999], our training does not require paired data between 2D images and 3D shapes, nor dense correspondence annotations in 3D data. This advantage allows us to leverage both 2D image datasets and 3D shape collections [Chang et al., 2015] and to synthesize objects of diverse shapes and texture.

Through extensive experiments, we show that VON produce more realistic image samples than recent 2D deep generative models. We also demonstrate many 3D applications that are enabled by our disentangled representation, including rotating an object, adjusting object shape and texture, interpolating between two objects in texture and shape space independently, and transferring the appearance of a real image to new objects and viewpoints.

## 2    Related Work

**GANs for 2D image synthesis.**    Since the invention of Generative Adversarial Nets (GANs) [Goodfellow et al., 2014], many researchers have adopted adversarial learning for various image synthesis tasks, ranging from image generation [Radford et al., 2016, Arjovsky et al., 2017, Karras et al., 2018], image-to-image translation [Isola et al., 2017, Zhu et al., 2017a], text-to-image synthesis [Zhang et al., 2017, Reed et al., 2016], and interactive image editing [Zhu et al., 2016, Wang et al., 2018], to classic vision and graphics tasks such as inpainting [Pathak et al., 2016] and super-resolution [Ledig et al., 2017]. Despite the tremendous progress made on 2D image synthesis, most of the above methods operate on 2D space, ignoring the 3D nature of our physical world. As a result, the lack of 3D structure inevitably limits some practical applications of these generative models. In contrast, we present an image synthesis method powered by a disentangled 3D representation.It allows a user to change the viewpoint easily, as well as to edit the object's shape or texture independently. Dosovitskiy et al. [2015] used supervised CNNs for generating synthetic images given object style, viewpoint, and color. We differ in that our aim is to produce objects with 3D geometry and natural texture without using labelled data.

**3D shape generation.**    There has been an increasing interest in synthesizing 3D shapes with deep generative models, especially GANs. Popular representations include voxels [Wu et al., 2016], point clouds [Gadelha et al., 2017b, Achlioptas et al., 2018], and octave trees [Tatarchenko et al., 2017]. Other methods learn 3D shape priors from 2D images [Rezende et al., 2016, Gadelha et al., 2017a]. Recent work also explored 3D shape completion from partial scans with deep generative models [Dai et al., 2017, Wang et al., 2017, Wu et al., 2018], including generalization to unseen object categories [Zhang et al., 2018]. Unlike prior methods that only synthesize untextured 3D shapes, our method learns to produce both realistic shapes and images. Recent and concurrent work has learned to infer both texture and 3D shapes from 2D images, represented as parametrized meshes [Kanazawa et al., 2018], point clouds [Tatarchenko et al., 2016], or colored voxels [Tulsiani et al., 2017, Sun et al., 2018b]. While they focus on 3D reconstruction, we aim to learn an unconditional generative model of shapes and images with disentangled representations of object texture, shape and pose.

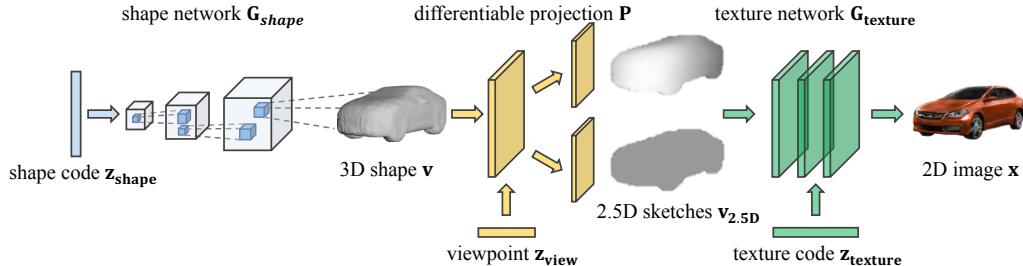

shape network $\mathbf{G}_{shape}$       differentiable projection $\mathbf{P}$       texture network $\mathbf{G}_{texture}$

shape code $\mathbf{z_{shape}}$       3D shape $\mathbf{v}$       2D image $\mathbf{x}$

2.5D sketches $\mathbf{v_{2.5D}}$

viewpoint $\mathbf{z_{view}}$       texture code $\mathbf{z_{texture}}$

Figure 2: **Our image formation model.** We first learn a shape generative adversarial network $G_{\text{shape}}$ that maps a randomly sampled shape code $\mathbf{z}_{\text{shape}}$ to a voxel grid $\mathbf{v}$. Given a sampled viewpoint $\mathbf{z}_{\text{view}}$, we project $\mathbf{v}$ to 2.5D sketches $\mathbf{v}_{2.5D}$ with our differentiable projection module $\mathcal{P}$. The 2.5D sketches $\mathbf{v}_{2.5D}$ include both the object's depth and silhouette, which help to bridge 3D and 2D data. Finally, we learn a texture network $\mathbf{x} = G_{\text{texture}}(\mathbf{v}_{2.5D}, \mathbf{z}_{\text{texture}})$ to add realistic, diverse texture to these 2.5D sketches, so that generated 2D images cannot be distinguished from real images by an image discriminator. The model is fully differentiable and trained end-to-end on both 2D and 3D data.

**Inverse graphics.** Motivated by the philosophy of "vision as inverse graphics" [Yuille and Kersten, 2006, Bever and Poeppel, 2010], researchers have made much progress in recent years on learning to invert graphics engines, many with deep neural networks [Kulkarni et al., 2015b, Yang et al., 2015, Kulkarni et al., 2015a, Tung et al., 2017, Shu et al., 2017]. In particular, Kulkarni et al. [2015b] proposed a convolutional inverse graphics network. Given an image of a face, the network learns to infer properties such as pose and lighting. Tung et al. [2017] extended inverse graphics networks with adversarial learning. Wu et al. [2017, 2018] inferred 3D shapes from a 2D image via 2.5D sketches and learned shape priors. Here we focus on a complementary problem—learning generative graphics networks via the idea of "graphics as inverse vision". In particular, we learn our generative model with recognition models that recover 2.5D sketches from generated images.

## 3 Formulation

Our goal is to learn an (implicit) generative model that can sample an image $x \in \mathbb{R}^{H \times W \times 3}$ from three factors: a shape code $\mathbf{z}_{\text{shape}}$, a viewpoint code $\mathbf{z}_{\text{view}}$, and a texture code $\mathbf{z}_{\text{texture}}$. The texture code describes the appearance of the object, which accounts for the object's albedo, reflectance, and environment illumination. These three factors are disentangled, conditionally independent from each other. Our model is category-specific, as the visual appearance of an object depends on the class. We further assume that all the codes lie in their own low-dimensional spaces. During training, we are given a 3D shape collection $\{\mathbf{v}_i\}_i^N$, where $\mathbf{v}_i \in \mathbb{R}^{W \times W \times W}$ is a binary voxel grid, and a 2D image collection $\{\mathbf{x}_j\}_j^M$, where $\mathbf{x}_j \in \mathbb{R}^{H \times W \times 3}$. Our model training requires *no* alignment between 3D and 2D data. We assume that every training image has a clean background and only contains the object of interest. This assumption makes our model focus on generating realistic images of the objects instead of complex backgrounds.

Figure 2 illustrates our model. First, we learn a 3D shape generation network that produces realistic voxels $\mathbf{v} = G_{\text{shape}}(\mathbf{z}_{\text{shape}})$ given a shape code $\mathbf{z}_{\text{shape}}$ (Section 3.1). We then develop a differentiable projection module $\mathcal{P}$ that projects a 3D voxel grid $\mathbf{v}$ into 2.5D sketches via $\mathbf{v}_{2.5D} = \mathcal{P}(\mathbf{v}, \mathbf{z}_{\text{view}})$, given a particular viewpoint $\mathbf{z}_{\text{view}}$ (Section 3.2). Next, we learn to produce a final image given the 2.5D sketches $\mathbf{v}_{2.5D}$ and a randomly sampled texture code $\mathbf{z}_{\text{texture}}$, using our texture synthesis network $\mathbf{x} = G_{\text{texture}}(\mathbf{v}_{2.5D}, \mathbf{z}_{\text{texture}})$ in Section 3.3. Section 3.4 summarizes our full model and Section 3.5 includes implementation details. Our entire model is differentiable and can be trained end-to-end.

During testing, we sample an image $x = G_{\text{texture}}(\mathcal{P}(G_{\text{shape}}(\mathbf{z}_{\text{shape}}), \mathbf{z}_{\text{view}}), \mathbf{z}_{\text{texture}})$ from latent codes $(\mathbf{z}_{\text{shape}}, \mathbf{z}_{\text{view}}, \mathbf{z}_{\text{texture}})$ via our shape network $G_{\text{shape}}$, texture network $G_{\text{texture}}$, and projection module $\mathcal{P}$.

### 3.1 Learning 3D Shape Priors

Our first step is to learn a category-specific 3D shape prior from large shape collections [Chang et al., 2015]. This prior depends on the object class but is conditionally independent of other factors such as

viewpoint and texture. To model the 3D shape prior and generate realistic shapes, we adopt the 3D Generative Adversarial Networks recently proposed by Wu et al. [2016].

Consider a voxelized 3D object collection $\{\mathbf{v}_i\}_i^N$, where $\mathbf{v}_i \in \mathbb{R}^{W \times W \times W}$. We learn a generator $G_{\text{shape}}$ to map a shape code $\mathbf{z}_{\text{shape}}$, randomly sampled from a Gaussian distribution, to a $W \times W \times W$ voxel grid. Simultaneously, we train a 3D discriminator $D_{\text{shape}}$ to classify a shape as real or generated. Both discriminator and generator contain fully volumetric convolutional and deconvolutional layers. We find that the original 3D-GAN [Wu et al., 2016] sometimes suffers from mode collapse. To improve the quality and diversity of the results, we use the Wasserstein distance of WGAN-GP [Arjovsky et al., 2017, Gulrajani et al., 2017]. Formally, we play the following minimax two-player game between $G_{\text{shape}}$ and $D_{\text{shape}}$: $\min_{G_{\text{shape}}} \max_{D_{\text{shape}}} \mathcal{L}_{\text{shape}}^{\text{GAN}}$*, where

$$\mathcal{L}_{\text{shape}} = \mathbb{E}_{\mathbf{v}}[D_{\text{shape}}(\mathbf{v})] - \mathbb{E}_{\mathbf{z}_{\text{shape}}}[D_{\text{shape}}(G_{\text{shape}}(\mathbf{z}_{\text{shape}}))]. \tag{1}$$

To enforce the Lipschitz constraint in Wasserstein GANs [Arjovsky et al., 2017], we add a gradient-penalty loss $\lambda_{\text{GP}} \mathbb{E}_{\tilde{\mathbf{v}}}[(\nabla_{\tilde{\mathbf{v}}} D_{\text{shape}}(\tilde{\mathbf{v}}) - 1)^2]$ to Eqn. 1, where $\tilde{\mathbf{v}}$ is a randomly sampled point along the straight line between a real shape and a generated shape, and $\lambda_{\text{GP}}$ controls the capacity of $D_{\text{shape}}$. Since binary data is often challenging to model using GANs, we also experiment with distance function (DF) representation [Curless and Levoy, 1996], which is continuous on the 3D voxel space. See Section 4.1 for quantitative evaluations.

### 3.2 Generating 2.5D Sketches

Given a synthesized voxelized shape $v = G_{\text{shape}}(\mathbf{z}_{\text{shape}})$, how can we connect it to a 2D image? Inspired by recent work on 3D reconstruction [Wu et al., 2017], we use 2.5D sketches [Barrow and Tenenbaum, 1978, Marr, 1982] to bridge the gap between 3D and 2D. This intermediate representation provides three main advantages. First, generating 2.5D sketches from a 3D voxel grid is straightforward, as the projection is differentiable with respect to both the input shape and the viewpoint. Second, 2D image synthesis from a 2.5D sketch can be cast as an image-to-image translation problem [Isola et al., 2017], where existing methods have achieved successes even without paired data [Zhu et al., 2017a]. Third, compared with alternative approaches such as colored voxels, our method enables generating images at a higher resolution.

Here we describe our differentiable module for projecting voxels into 2.5D sketches. The inputs to this module are the camera parameters and 3D voxels. The value of each voxel stores the probability of it being present. To render the 2.5D sketches from the voxels under a perspective camera, we first generate a collection of rays, each originating from the camera's center and going through a pixel's center in the image plane. To render the 2.5D sketches, we need to calculate whether a given ray would hit the voxels, and if so, the corresponding depth value of that ray. To this end, we first sample a collection of points at evenly spaced depth along each ray. Next, for each point, we calculate the probability of hitting the input voxels using a differentiable trilinear interpolation [Jaderberg et al., 2015] of the input voxels. Similar to Tulsiani et al. [2017], we then calculate the expectation of visibility and depth along each ray. Specifically, given a ray $R$ with $N$ samples $R_1$, $R_2$, ... , $R_N$ along its path, we calculate the visibility (silhouette) as the expectation of the ray hitting the voxels: $\sum_{j=1}^{N} \prod_{k=1}^{j-1}(1 - R_k) R_j$. Similarly, the expected depth can be calculated as $\sum_{j=1}^{N} d_j \prod_{k=1}^{j-1}(1 - R_k) R_j$, where $d_j$ is the depth of the sample $R_j$. This process is fully differentiable since the gradients can be back-propagated through both the expectation calculation and the trilinear interpolation.

**Viewpoint estimation.** Our two-dimensional viewpoint code $\mathbf{z}_{\text{view}}$ encodes camera elevation and azimuth. We sample $\mathbf{z}_{\text{view}}$ from an empirical distribution $p_{\text{data}}(\mathbf{z}_{\text{view}})$ of the camera poses from the training images. To estimate $p_{\text{data}}(\mathbf{z}_{\text{view}})$, we first render the silhouettes of several candidate 3D models under uniformly sampled camera poses. For each input image, we compare its silhouette to the rendered 2D views and choose the pose with the largest Intersection-over-Union value. More details can be found in the supplement.

## 3.3 Learning 2D Texture Priors

Next, we learn to synthesize realistic 2D images given projected 2.5D sketches that encode both the viewpoint and the object shape. In particular, we learn a texture network $G_{\text{texture}}$ that takes a randomly sampled texture code $\mathbf{z}_{\text{texture}}$ and the projected 2.5D sketches $\mathbf{v}_{\text{2.5D}}$ as input, and produces a 2D image $\mathbf{x} = G_{\text{texture}}(\mathbf{v}_{\text{2.5D}}, \mathbf{z}_{\text{texture}})$. This texture network needs to model both object texture and environment illumination, as well as the differentiable rendering equation [Kajiya, 1986]. Fortunately, this mapping problem can be cast as an unpaired image-to-image translation problem [Zhu et al., 2017a, Yi et al., 2017, Liu et al., 2017]. We adopt recently proposed cycle-consistent adversarial networks (CycleGAN) [Zhu et al., 2017a] as our baseline. Later, we relax the one-to-one mapping restriction in CycleGAN to handle one-to-many mappings from 2.5D sketches to 2D images.

Here we introduce two encoders $E_{\text{texture}}$ and $E_{\text{2.5D}}$ to estimate a texture code $\mathbf{z}_{\text{texture}}$ and 2.5D sketches $\mathbf{v}_{\text{2.5D}}$ from a real image $\mathbf{x}$. We train $G_{\text{texture}}$, $E_{\text{texture}}$, and $E_{\text{2.5D}}$ jointly with adversarial losses [Goodfellow et al., 2014] and cycle-consistency losses [Zhu et al., 2017a, Yi et al., 2017]. We use the following adversarial loss on the final generated image:

$$\mathcal{L}_{\text{image}}^{\text{GAN}} = \mathbb{E}_{\mathbf{x}}[\log D_{\text{image}}(\mathbf{x})] + \mathbb{E}_{(\mathbf{v}_{\text{2.5D}}, \mathbf{z}_{\text{texture}})}[\log(1 - D_{\text{image}}(G_{\text{texture}}(\mathbf{v}_{\text{2.5D}}, \mathbf{z}_{\text{texture}})))], \quad (2)$$

where $D_{\text{image}}$ learns to classify real and generated images. We apply the same adversarial loss for 2.5D sketches $\mathbf{v}_{\text{2.5D}}$:

$$\mathcal{L}_{\text{2.5D}}^{\text{GAN}} = \mathbb{E}_{\mathbf{v}_{\text{2.5D}}}[\log D_{\text{2.5D}}(\mathbf{v}_{\text{2.5D}})] + \mathbb{E}_{\mathbf{x}}[\log(1 - D_{\text{2.5D}}(E_{\text{2.5D}}(\mathbf{x})))], \quad (3)$$

where $D_{\text{2.5D}}$ aims to distinguish between 2.5D sketches $\mathbf{v}_{\text{2.5D}}$ and estimated 2.5D sketches $E_{\text{2.5D}}(\mathbf{x})$ from a real 2D image. We further use cycle-consistency losses [Zhu et al., 2017a] to enforce the bijective relationship between the two domains:

$$\mathcal{L}_{\text{2.5D}}^{\text{cyc}} = \lambda_{\text{2.5D}}^{\text{cyc}} \mathbb{E}_{(\mathbf{v}_{\text{2.5D}}, \mathbf{z}_{\text{texture}})}\left[\|E_{\text{2.5D}}(G_{\text{texture}}(\mathbf{v}_{\text{2.5D}}, \mathbf{z}_{\text{texture}})) - \mathbf{v}_{\text{2.5D}}\|_1\right]$$

$$\text{and} \quad \mathcal{L}_{\text{image}}^{\text{cyc}} = \lambda_{\text{image}}^{\text{cyc}} \mathbb{E}_{\mathbf{x}}\left[\|G_{\text{texture}}(E_{\text{2.5D}}(\mathbf{x}), E_{\text{texture}}(\mathbf{x})) - \mathbf{x}\|_1\right], \quad (4)$$

where $\lambda_{\text{image}}^{\text{cyc}}$ and $\lambda_{\text{2.5D}}^{\text{cyc}}$ control the importance of each cycle loss. TThe texture encoder $E_{\text{texture}}$ and 2.5D sketch encoder $E_{\text{2.5D}}$ serve as recognition models that recover the texture and 2.5D representation from a 2D image.

**One-to-many mappings.** Prior studies [Isola et al., 2016, Mathieu et al., 2016] have found that latent codes are often ignored in conditional image generation due to the assumption of a one-to-one mapping; vanilla CycleGAN also suffers from this problem based on our experiments. To address this, we introduce a latent space cycle-consistency loss to encourage $G_{\text{texture}}$ to use the texture code $\mathbf{z}_{\text{texture}}$:

$$\mathcal{L}_{\text{texture}}^{\text{cyc}} = \lambda_{\text{texture}}^{\text{cyc}} \mathbb{E}_{(\mathbf{v}_{\text{2.5D}}, \mathbf{z}_{\text{texture}})}[\|E_{\text{texture}}(G_{\text{texture}}(\mathbf{v}_{\text{2.5D}}, \mathbf{z}_{\text{texture}})) - \mathbf{z}_{\text{texture}}\|_1], \quad (5)$$

where $\lambda_{\text{texture}}^{\text{cyc}}$ controls its importance. Finally, to allow sampling at test time, we add a Kullback–Leibler (KL) loss on the $z$ space to force $E_{\text{texture}}(\mathbf{x})$ to be close to a Gaussian distribution:

$$\mathcal{L}_{\text{KL}} = \lambda_{\text{KL}} \mathbb{E}_{\mathbf{x}}\left[\mathcal{D}_{\text{KL}}(E_{\text{texture}}(\mathbf{x})\|\mathcal{N}(0, I))\right], \quad (6)$$

where $\mathcal{D}_{\text{KL}}(p\|q) = -\int_{\mathbf{z}} p(\mathbf{z}) \log \frac{p(\mathbf{z})}{q(\mathbf{z})} dz$ and $\lambda_{\text{KL}}$ is its weight. We write the final texture loss as

$$\mathcal{L}_{\text{texture}} = \underbrace{\mathcal{L}_{\text{image}}^{\text{GAN}} + \mathcal{L}_{\text{2.5D}}^{\text{GAN}}}_{\text{Adversarial losses}} + \underbrace{\mathcal{L}_{\text{image}}^{\text{cyc}} + \mathcal{L}_{\text{2.5D}}^{\text{cyc}} + \mathcal{L}_{\text{texture}}^{\text{cyc}}}_{\text{Cycle-consistency losses}} + \underbrace{\mathcal{L}_{\text{KL}}}_{\text{KL loss}}. \quad (7)$$

Note that the latent space reconstruction loss $\mathcal{L}_{\text{texture}}^{\text{cyc}}$ has been explored in unconditional GANs [Chen et al., 2016] and image-to-image translation [Zhu et al., 2017b, Almahairi et al., 2018]. Here we use this loss to learn one-to-many mappings from unpaired data.

## 3.4 Our Full Model

Our full objective is

$$\underset{(G_{\text{shape}}, G_{\text{texture}}, E_{\text{2.5D}}, E_{\text{texture}})}{\arg\min} \underset{(D_{\text{shape}}, D_{\text{texture}}, D_{\text{2.5D}})}{\arg\max} \lambda_{\text{shape}} \mathcal{L}_{\text{shape}} + \mathcal{L}_{\text{texture}}, \quad (8)$$

where $\lambda_{\text{shape}}$ controls the relative weight of shape and texture loss functions. We compare our visual object networks against 2D deep generative models in Section 4.1.

## 3.5 Implementation Details

**Shape networks.** For shape generation, we adopt the 3D-GAN architecture from Wu et al. [2016]. In particular, the discriminator $D_{\text{shape}}$ contains 6 volumetric convolutional layers and the generator $G_{\text{shape}}$ contains 6 strided-convolutional layers. We remove the batch normalization layers [Ioffe and Szegedy, 2015] in the $G_{\text{shape}}$ as suggested by the WGAN-GP paper [Gulrajani et al., 2017].

**Texture networks.** For texture generation, we use the ResNet encoder-decoder [Zhu et al., 2017a, Huang et al., 2018] and concatenate the texture code $\mathbf{z}_{\text{texture}}$ to intermediate layers in the encoder. For the discriminator, we use two-scale PatchGAN classifiers [Isola et al., 2017, Zhu et al., 2017a] to classify overlapping patches as real or fake. We use a least square objective as in LS-GAN [Mao et al., 2017] for stable training. We use ResNet encoders [He et al., 2015] for our $E_{\text{texture}}$ and $E_{2.5D}$.

**Differentiable projection module.** We assume the camera is at a fixed distance of 2m to the object's center and use a focal length of 50mm (35mm film equivalent). The resolution of the rendered sketches are $128 \times 128$, and we sample 128 points evenly along each camera ray. We also assume no in-plane rotation, that is, no tilting in the image plane. We implement a custom CUDA kernel for sampling along the projection rays and calculating the stop probabilities.

**Training details.** We train our models on $128 \times 128 \times 128$ shapes (voxels or distance function) and $128 \times 128 \times 3$ images. During training, we first train the shape generator $G_{\text{shape}}$ on 3D shape collections and then train the texture generator $G_{\text{texture}}$ given ground truth 3D shape data and image data. Finally, we fine-tune both modules together. We sample the shape code $\mathbf{z}_{\text{shape}}$ and texture code $\mathbf{z}_{\text{texture}}$ from the standard Gaussian distribution $N(0, I)$, with the code length $|\mathbf{z}_{\text{shape}}| = 200$ and $|\mathbf{z}_{\text{texture}}| = 8$. The entire training usually takes two to three days. For hyperparameters, we set $\lambda_{\text{KL}} = 0.05$, $\lambda_{\text{GP}} = 10$, $\lambda_{\text{image}}^{\text{cyc}} = 10$, $\lambda_{2.5D}^{\text{cyc}} = 25$, $\lambda_{\text{texture}}^{\text{cyc}} = 1$, and $\lambda_{\text{shape}} = 0.05$. We use the Adam solver [Kingma and Ba, 2015] with a learning rate of 0.0002 for shape generation and 0.0001 for texture generation.

We observe that the texture generator $G_{\text{texture}}$ sometimes introduces the undesirable effect of changing the shape of the silhouette when rendering 2.5D sketches $\mathbf{v}_{2.5D}$ (i.e., depth and mask). To address this issue, we explicitly mask the generated 2D images with the silhouette from $\mathbf{v}_{2.5D}$: i.e., $G_{\text{texture}}(\mathbf{v}_{2.5D}, \mathbf{z}_{\text{texture}}) = \text{mask} \cdot g_{\text{texture}}(\text{depth}) + (1 - \text{mask}) \cdot \mathbf{1}$, where $\mathbf{1}$ is the background white color and the generator $g_{\text{texture}}$ synthesizes an image given a depth map. Similarly, we reformulate $E_{2.5D}(x) = (e_{2.5D}(x) \cdot \text{mask}_{\text{gt}}, \text{mask}_{\text{gt}})$, where the encoder $e_{2.5D}$ only predicts depth, and the input object mask is used. In addition, we add a small mask consistency loss $||e_{2.5D}(\mathbf{x}) - \text{mask}_{\text{gt}}||_1$ to encourage the predicted depth map to be consistent with the the object mask. As our training images have clean background, we can estimate the object mask with a simple threshold.

## 4 Experiments

We first compare our visual object networks (VON) against recent 2D GAN variants on two datasets. We evaluate the results using both a quantitative metric and a qualitative human perception study. We then perform an ablation study on the objective functions of our shape generation network. Finally, we demonstrate several applications enabled by our disentangled 3D representation. The full results and datasets can be found at our website. Please find our implementation at GitHub.

### 4.1 Evaluations

**Datasets.** We use ShapeNet [Chang et al., 2015] for learning to generate 3D shapes. ShapeNet is a large shape repository of 55 object categories. Here we use the chair and car categories, which has $6,777$ and $3,513$ CAD models respectively. For 2D datasets, we use the recently released Pix3D dataset to obtain $1,515$ RGB images of chairs alongside with their silhouettes [Sun et al., 2018a], with an addition of $448$ clean background images crawled from Google image search. We also crawled $2,605$ images of cars.

**Baselines** We compare our method to three popular GAN variants commonly used in the literature: DCGAN with the standard cross-entropy loss [Goodfellow et al., 2014, Radford et al., 2016], LSGAN [Mao et al., 2017], and WGAN-GP [Gulrajani et al., 2017]. We use the same DCGAN-like generator and discriminator architectures for all three GAN models. For WGAN-GP, we replace the BatchNorm by InstanceNorm [Ulyanov et al., 2016] in the discriminator, and we train the discriminator 5 times per generator iteration.

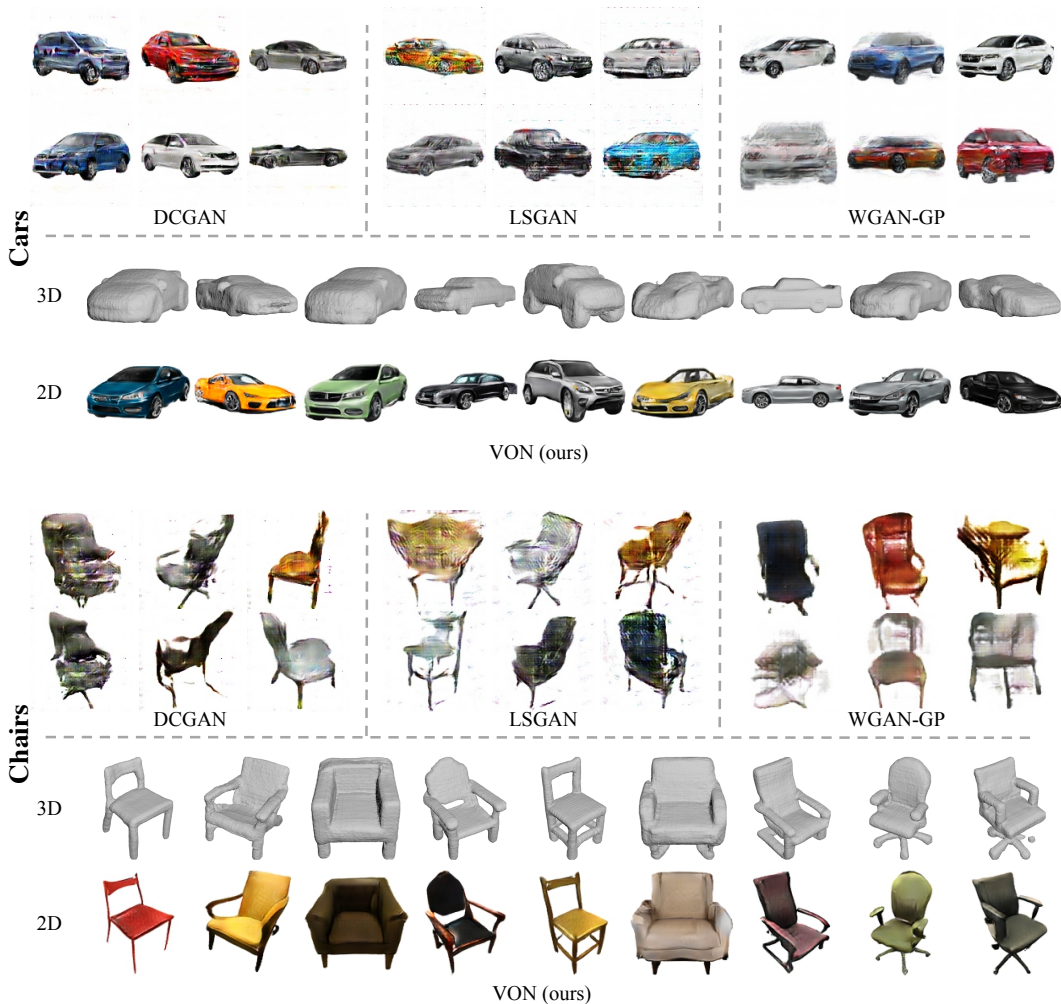

Figure 3: Qualitative comparisons between 2D GAN models and VON: we show samples from DCGAN [Radford et al., 2016], LSGAN [Mao et al., 2017], WGAN-GP [Gulrajani et al., 2017], and our VON. For our method, we show both 3D shapes and 2D images. Note that VON is trained on unpaired 3D shapes and 2D images, while DCGAN, LSGAN and WGAN-GP are trained only on 2D images. The learned 3D prior helps our model produce better samples. (Top: cars; bottom: chairs.)

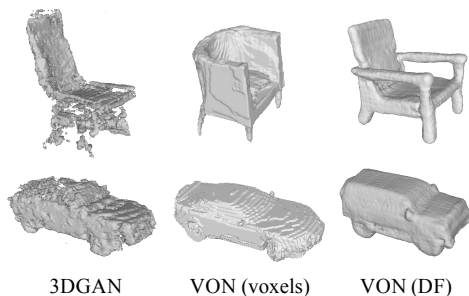

3DGAN          VON (voxels)          VON (DF)

Figure 4: Sampled 3D shapes: from left to right: 3DGAN [Wu et al., 2016], VON on voxels, VON on distance functions (DF). Our model produce more natural 3D shapes. OUr model produces samples with higher quality.

|  | 3D-GAN (voxels) | VON (voxels) |
|---|---|---|
| Cars | 3.021 | **0.021** |
| Chairs | 2.598 | **0.082** |
|  | 3D-GAN (DF) | VON (DF) |
| Cars | 3.896 | **0.002** |
| Chairs | 1.790 | **0.006** |

Table 3: Quantitative comparisons on shape generation: Fréchet Inception Distances (FID) between real shapes and shapes generated by 3D-GAN [Wu et al., 2016] and our shape network, both on voxels and distance function representation (DF). Our model achieves better results regarding FID.

|          | Car   | Chair |
|----------|-------|-------|
| DCGAN    | 130.5 | 225.0 |
| LSGAN    | 171.4 | 225.3 |
| WGAN-GP  | 123.4 | 184.9 |
| VON (voxels) | **81.6** | 58.0 |
| VON (DF) | **83.3** | **51.8** |

Table 1: Fréchet Inception Distances [Heusel et al., 2017] between real images and images generated by DCGAN, LSGAN, WGAN-GP, our VON (voxels), and our VON (DF). DF denotes distance function representations.

|          | Car    | Chair |
|----------|--------|-------|
| DCGAN    | 72.2%  | 90.3% |
| LSGAN    | 78.7%  | 92.4% |
| WGAN-GP  | 63.0 % | 89.1% |

Table 2: Human preferences on images generated by DCGAN [Radford et al., 2016], LSGAN [Mao et al., 2017], WGAN-GP [Gulrajani et al., 2017] *vs*. our VON (DF). Each number shows the percentage of human subjects who prefer our method to the baseline method.

**Metrics.** To evaluate the image generation models, we calculate the Fréchet Inception Distance between generated images and real images, a metric highly correlated to human perception [Heusel et al., 2017, Lucic et al., 2018]. Each set of images are fed to the Inception network [Szegedy et al., 2015] trained on ImageNet [Deng et al., 2009], and the features from the layer before the last fully-connected layer are used to calculate the Fréchet Inception Distance.

Second, we sample 200 pairs of generated images from the VON and the state-of-the-art models (DCGAN, LSGAN, and WGAN-GP), and show each pair to five subjects on Amazon MTurk. The subjects are asked to choose a more realistic result within the pair.

**Results** Our VON consistently outperforms the 2D generative models. In particular, Table 1 shows that our results have the smallest Fréchet Inception Distance; in Table 2, $74\% - 85\%$ of the responses preferred our results. This performance gain demonstrates that the learned 3D prior helps synthesize more realistic images. See Figure 3 for a qualitative comparison between these methods.

**Analysis of shape generation.** For shape generation, we compare our method against the prior 3D-GAN work by Wu et al. [2016] on both voxel grids and distance function representation. 3D-GAN uses the same architecture but trained with a cross-entropy loss. We evaluate the shape generation models using the Fréchet Inception Distance (FID) between the generated and real shapes. To extract statistics for each set of generated/real shapes, we train ResNet-based 3D shape classifiers [He et al., 2015] on all 55 classes of shapes from ShapeNet; classifiers are trained separately on voxels and distance function representations. We extract the features from the layer before the last fully-connected layer. Table 3 shows that our method achieves better results regarding FID. Figure 4a shows that the Wasserstein distance increases the quality of the results. As we use different classifiers for voxels and distance functions, the Fréchet Inception Distance is not comparable across representations.

## 4.2 Applications

We apply our visual object networks to several 3D manipulation applications, not possible by previous 2D generative models [Goodfellow et al., 2014, Kingma and Welling, 2014].

**Changing viewpoints.** As our VON first produces a 3D shape, we can project the shape to the image plane given different viewpoints $\mathbf{z}_{\text{view}}$ while keeping the same shape and texture code. Figure 1c and Figure 5a show a few examples.

**Shape and texture editing.** With our learned disentangled 3D representation, we can easily change only the shape code or the texture code, which allows us to edit the shape and texture separately. See Figure 1c and Figure 5a for a few examples.

**Disentangled interpolation.** Given our disentangled 3D representation, we can choose to interpolate between two objects in different ways. For example, we can interpolate objects in shape space $\alpha \mathbf{z}_{\text{shape}}^1 + (1 - \alpha)\mathbf{z}_{\text{shape}}^2$ with the same texture, or in the texture space $\alpha \mathbf{z}_{\text{texture}}^1 + (1 - \alpha)\mathbf{z}_{\text{texture}}^2$ with the same shape, or both, where $\alpha \in [0, 1]$. Figure 5c shows linear interpolations in the latent space.

**Example-based texture transfer.** We can infer the texture code $\mathbf{z}_{\text{texture}}$ from a real image $\mathbf{x}$ with the texture encoder $\mathbf{z}_{\text{texture}} = E_{\text{texture}}(\mathbf{x})$, and apply the code to new shapes. Figure 6 shows texture transfer results on cars and chairs using real images and generated shapes.

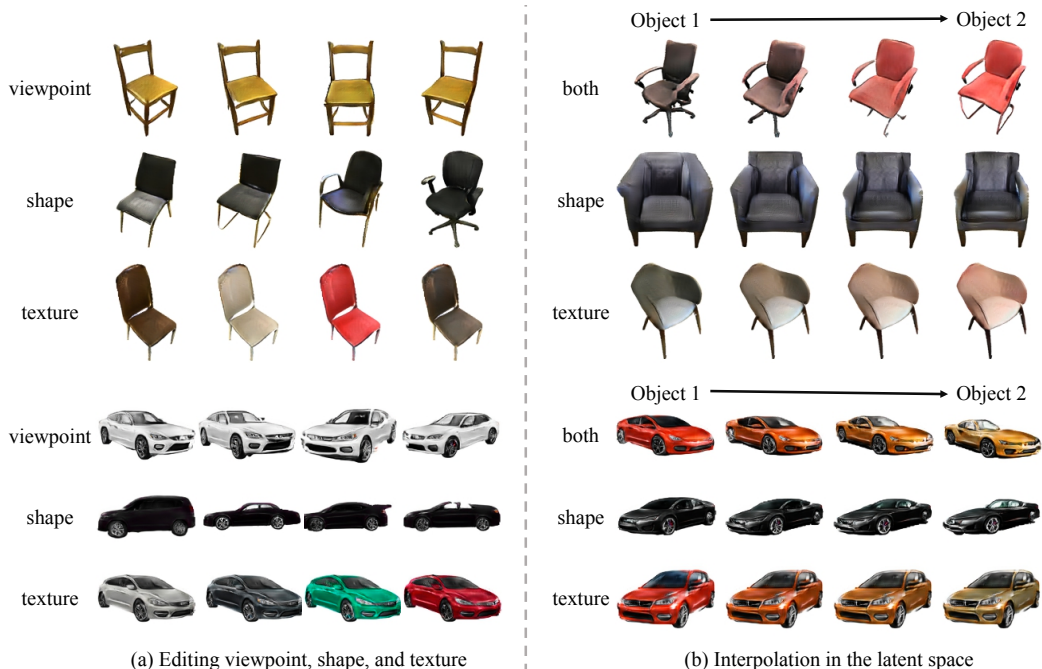

(a) Editing viewpoint, shape, and texture      (b) Interpolation in the latent space

Figure 5: 3D-aware applications: Our visual object networks allow several 3D applications such as (a) changing the viewpoint, texture, or shape independently, and (b) interpolating between two objects in shape space, texture space, or both. None of them can be achieved by previous 2D GANs.

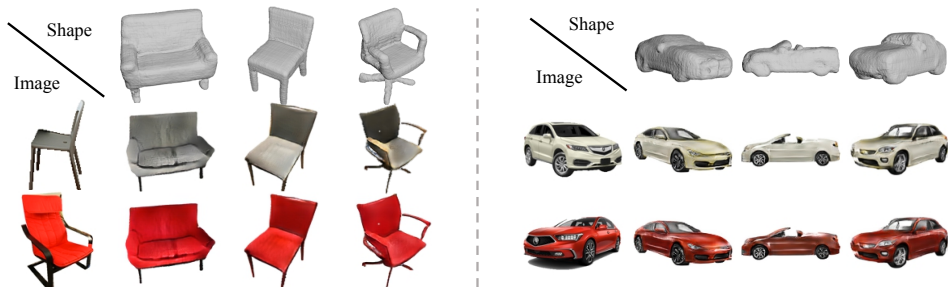

Figure 6: Given a real input image, we synthesize objects with similar texture using the inferred texture code. The same texture is transferred to different shapes and viewpoints.

## 5 Discussion

In this paper, we have presented visual object networks (VON), a fully differentiable 3D-aware generative model for image and shape synthesis. Our key idea is to disentangle the image generation process into three factors: shape, viewpoint, and texture. This disentangled 3D representation allows us to learn the model from both 3D and 2D visual data collections under an adversarial learning framework. Our model synthesizes more photorealistic images compared to existing 2D generative models; it also enables various 3D manipulations that are not possible with prior 2D methods.

In the future, we are interested in incorporating coarse-to-fine modeling [Karras et al., 2017] for producing shapes and images at a higher resolution. Another interesting direction to explore is to disentangle texture further into lighting and appearance (e.g., albedo), which could improve the consistency of appearance across different viewpoints and lighting conditions. Finally, as we do not have large-scale 3D geometric data for entire scenes, our current method only works for individual objects. Synthesizing natural scenes is also a meaningful next step.

**Acknowledgements** This work is supported by NSF #1231216, NSF #1447476, NSF #1524817, ONR MURI N00014-16-1-2007, Toyota Research Institute, Shell, and Facebook. We thank Xiuming Zhang, Richard Zhang, David Bau, and Zhuang Liu for valuable discussions.

## Footnotes

*For notation simplicity, we denote $\mathbb{E}_{\mathbf{v}} \triangleq \mathbb{E}_{\mathbf{v} \sim p_{\text{data}}(\mathbf{v})}$ and $\mathbb{E}_{\mathbf{z}_{\text{shape}}} \triangleq \mathbb{E}_{\mathbf{z}_{\text{shape}} \sim p_{\text{data}}(\mathbf{z}_{\text{shape}})}$.

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
