[Reviews · NeurIPS 2018]

Reviewer 1



The paper presents a generative model called Visual Object Networks (VQN), which models 3D shape and 2D texture. Loss for shapes L_shape and loss for textures L_texture are added. L_shape consists of loss for GAN and 3D geometry loss. L_texture consists of GAN loss and cycle loss from images and textures. Empirical evaluation shows the superiority to other existing models in several measures such as inception distance, human preference, and log-likelihood. Overall, the paper is well-written and well-organized. The only concern is that the proposed method is rather straightforward, and does not convey much insight to the NIPS society. Handling textures and shapes separately is also a limitation from the algorithmic perspective. Finally, more elaborate comparison to other models (including Generative Query Networks) may improve the paper.

Reviewer 2



This paper describes a generative model for image formation, with disentangled latent parameters for shape, viewpoint and texture. This is in keeping with the vision as an inverse graphics problem, where image generation is formulated as a parameter search in model space, that when rendered, produces the given image. The difference between the rendered image and the original image is used to train the model. Using inverse graphics as inspiration, this paper learns the following models: 1. An voxel generator that can map the latent 3D shape code to a voxellized 3D shape 2.A differentiable projection module that converts the output of 1 to a 2.5D sketch (depth map) and a silhouette mask, conditional on a latent representation of the required viewpoint 3.A texture generator, that can map the output of 2 to a realistic textured image, conditional on a latent representation of the required texture 4.A 2.5D sketch encoder that can map a 2D image to a 2.5D sketch 5.A texture encoder that maps the texture of the object in a 2D image into a texture latent code The models are learnt adversarially using GANs, and without the need for paired image and shape data using the now common cycle consistency constraints. The training of the GANs are improved using the recently described WGAN that enforces the Lipshitz constraint with a gradient normalizing term, which leads to a better optimization landscape and prevents mode collapse. Results on car and chair datasets indicate qualitative and quantitative (Frechet Inception distance) improvements in image generation by varying pose, texture and shape, compared to competing techniques. The trained models and disentangled latent parameters of shape, texture and viewpoint allow a mix and match between the same. Quality This work demonstrates a generative model of image formation with disentangled factors that can be trained without paired training data between factors. The work is well-reasoned and represents an advance in tying together the factors responsible for image formation, and how to train generative models to train each of those, end-to-end. Clarity The paper is well written, with the motivation GAN losses well-explained. The following points require the writers’ attention: Line 26 - you mean virtual reality and not visual reality Line 103 - do you classify individual voxels, and not the whole voxel shape as real/fake? More explanation required. Section 3.2 - More details required on the projection module, perhaps a diagram. And how is it differentiable? Figure 5 (b) Texture transfer does not seem to work as well in the first row, where the white car’s texture is transferred. The two cars to whom the texture has been transferred look more grey/silver than white. This could be a consequence of lighting change. Perhaps lighting is another factor that requires consideration as a factor in a disentangled generative model of image formation. Figure 5 (c) It would clarify things if you show the 2 objects you are interpolating between. Results on the 3rd row are not very impressive - the front of the car seems mashed up, and not a very smooth transition between shapes. An explanation in the text is required. Why do you not use the W-GAN loss in the loss functions for 2.5D GAN and texture GAN, when you use it for the shape GAN? In addition to the applications listed in 4.3, it would be useful to see if you can apply this to objects for which the 3D model does not exist during training. Can you go from 2D images (from 1 or 2 viewpoints) of a new object (unseen during training), and go from the 2D images to the 3D shape, and then generate 2D images of the object from a new viewpoint? This would be a test of how well your model is able to generalize over shapes. Originality This paper describes an incremental advance in the development of generative models of image formation. It has enough originality to warrant publication at NIPS. Significance The inverse graphics style generative model with disentangled factors for image formation, that can be trained without paired training data between the factors is a fundamental development in the field of Computer Vision, and has the potential for significant advances in applications like virtual reality, augmented reality and robotics. This paper presents a well written and comprehensive treatment of an incremental improvement in this direction.

Reviewer 3



The paper proposes a novel CNN based architecture generating 2D images of 3D objects. In contrast to existing GAN methods, the proposed approach combines two independent generators: one for 3D shape and another for texture. The first component is a CNN mapping a low dimensional code to 3D volumetric shape. The second component is a CNN mapping 2.5D sketch (depth + silhouette) and a low-dim texture code to the final 2D image. The two components are connected by 3D to 2.5D translator, conditioned by viewing angle, having simple algorithmic solution. The 3D generator and the texture generator are learned from training data, 3D objects and 2D images, which do need to be aligned. Learning of 3D shape generator is based on 3D-GAN [Wu et al 2016] with a slight improvement. Learning of the texture generator uses a technique similar to image-to-image translation learned by CycleGANs [Zhu et al 2017]. The paper is clearly written up to a few typos and points mentioned below. The main novelty is in the proposed CNN-based architecture for image generation which disentangles 3D shape and texture and which can be learned from unaligned data. The experimental evaluation is sufficiently convincing. Detailed comments: - Learning of the full model, equ (9), leads to a complicated min-max problem with a complicated objective. It would be useful to see at least an outline of the optimization scheme used. It is also not clear what is the gain from solving (9) as compared to learning both texture and 3d shape models separately. - The 3d generator is claimed to be learned by Wasserstein GAN, however, the objective function (1) corresponds to the original GAN [Goodfellow 2014]. It is not clear why. - Description of the probabilistic ray-voxel intersection algorithm (line 135-140) is not fully clear. It requires reference or more details. - The shape code (line 201) is claimed to be sampled from Gaussian distribution, however, it is not clear why it should follow this distribution. The Gaussian distribution should be probably enforced when learning the 3d shape generator but it is not the case according to the formulas. - In the ablation study, the texture generator is evaluated by means of Frechet Inception Distance. In addition to this, one could measure (using e.g. Hamming distance) to what extent the texture generator can copy silhouette correctly. Typos: - line 88: $z_{shape}$ -> $z-{view}$ - line 173: $G_{shape}$ -> $G_{texture}$ - line 242: " loss, We"